ESR2 gene variants (rs1256049, rs4986938, and rs1256030) and their association with breast cancer risk

Gallegos-Arreola Martha Patricia 1 marthapatriciagallegos08@gmail.com
Zúñiga-González Guillermo M. 2
Figuera Luis E. 1
Puebla-Pérez Ana María 3
Márquez-Rosales María Guadalupe 1
Gómez-Meda Belinda Claudia 4
Rosales-Reynoso Mónica Alejandra 2
1 División de Genética, Centro de Investigación Biomédica de Occidente (CIBO), Instituto Mexicano del Seguro Social (IMSS) , Guadalajara, Jalisco , México
2 División de Medicina Molecular, Centro de Investigación Biomédica de Occidente (CIBO), Instituto Mexicano del Seguro Social (IMSS) , Guadalajara, Jalisco , México
3 Laboratorio de Inmunofarmacología, Departamento de Farmacología, Centro Universitario de Ciencias Exactas e Ingenierías, Universidad de Guadalajara , Guadalajara, Jalisco , México
4 Departamento de Biología Molecular y Genómica, Instituto de Genética Humana “Dr. Enrique Corona Rivera”, Centro Universitario de Ciencias de la Salud, Universidad de Guadalajara , Guadalajara, Jalisco , México
Ozdag Sevgili Hilal
Electronic publication date: 2022 May 10
Publication date: 2022
Volume: 10
Electronic Location ID: e13379
Received 2022 Feb 16; Accepted 2022 Apr 13
Copyright: © 2022 Gallegos-Arreola et al.
Copyright year: 2022
Copyright holder: Gallegos-Arreola et al.
License: This is an open access article distributed under the terms of the Creative Commons Attribution License, which permits unrestricted use, distribution, reproduction and adaptation in any medium and for any purpose provided that it is properly attributed. For attribution, the original author(s), title, publication source (PeerJ) and either DOI or URL of the article must be cited.
License URL: https://creativecommons.org/licenses/by/4.0/

Keywords: Variants, ESR2, Breast cancer, rs1256049, rs4986938, rs1256030, Polymorphisms

Funding: FIS/IMSS/PROT/G17/1661 grants This research was financially supported through FIS/IMSS/PROT/G17/1661 grants. The funders had no role in study design, data collection and analysis, decision to publish, or preparation of the manuscript.

==============================
Background

Variants of the estrogen receptor b (ESR2) gene have been associated with different types of cancer. However, these associations have been inconsistent. We genotyped the ESR2 variants (rs1256049, rs4986938, and rs1256030) in breast cancer (BC) patients and in healthy women.

Results

The variants rs1256049 and rs4986938 in the ESR2 gene were not associated with risk susceptibility in BC patients. However, the rs1256030 variant had an association as a risk factor for BC patients when compared with controls and BC patients for the TT genotype (odds ratio (OR) 1.86, 95% confidence intervals (CI) [1.05–3.28], p = 0.042). In addition, differences were observed in patients and controls carrying the TT genotype under 50 years of age (OR 1.85, 95% CI [1.05–3.27], p = 0.043). Thus, evident differences showed the rs1256030 variant in patients with TT, TC, and TC+TT genotypes with: (1) Stage IV (OR 1.60, 95% CI [1.06–2.54], p = 0.033), and (2) Luminal A (OR 1.60, 95% CI [0.47–0.21], p = 0.041), as well as in BC carriers of the TT genotype with indices of cellular proliferative (Ki-67) elevated (>20%) and overweight (OR 1.67, 95% CI [0.85–3.28], p = 0.041), respectively. In BC HER2 with lymph node metastasis, the TT genotype was a protective factor (OR 0.38, 95% CI [0.18–0.78], p = 0.005). The identification of haplotypes included two common GAT as risk factors (OR 3.1, 95% CI [1.31–7.72], p = 0.011) and GGC as a protective factor (OR 0.7, 95% CI [0.60–0.97], p = 0.034). The haplogenotype GGGATC was a risk factor (OR 2.5, 95% CI [1.28–5.0], p = 0.008).

Conclusion

The variant rs1256030 (TT) of the ESR2 gene and haplotype GAT were associated with susceptibility to BC as risk factors in this sample from the Mexican population.

Introduction

Breast cancer (BC) is the most common type of cancer among women in the world (Sung et al., 2020). In Mexico, BC is one of the major causes of mortality, and it is increasingly detected in young women (Sung et al., 2020; Shoemaker et al., 2018). The gradual accumulation of epigenetic and genetic events that occur in ductal and lobular normal breast cells can transform into malignant tumor cells (Sung et al., 2020; Shoemaker et al., 2018; Gallegos et al., 2017). Estrogens are important hormones that regulate the development of the mammary glands via an estrogen receptor (ER), and they are often significantly overexpressed in BC (Gallicchio et al., 2006; Maguire et al., 2005; Yu et al., 2011; Karlsson et al., 2016; Ghali et al., 2018). ER (known as alpha (ERα) and beta (ERβ)) is a nuclear receptor formed by a protein family that functions as transcriptional regulators activated by steroid hormones (Paterni et al., 2014; Karlsson et al., 2016). ERα and ERβ have antagonistic functions regarding cell proliferation, while ERα promotes and ERβ inhibits cell proliferation (Maguire et al., 2005; Treeck et al., 2009; Yu et al., 2011; Paterni et al., 2014; Karlsson et al., 2016). It has been described that the estrogen receptor β gene (ESR2) has GC-rich regions that are target sites for estrogen-responsive elements and that variants of the ESR2 gene have been associated with BC (Zhao, Dahlman-Wright & Gustafsson, 2010). ERs are often significantly overexpressed in many cancers, such as mammary adenocarcinoma, and specifically the lack of expression of the ESR2 gene is closely related to the occurrence and progression of cholangiocarcinoma (Haldosén, Zhao & Dahlman-Wright, 2014; Mahdavipour et al., 2017; Zhang et al., 2018; Carrillo et al., 2019). Approximately 75% of primary BC patients express ER and more than half of these cancerous cells express progesterone receptors. In postmenopausal women with BC, estrogen has been associated with an increased risk (Carrillo et al., 2019). The ESR2 gene is located on chromosome 14q22-24 and has eight exons; the promotor and first intron contain regulatory sequences of other introns (Haldosén, Zhao & Dahlman-Wright, 2014; Mahdavipour et al., 2017; Zhang et al., 2018). Several variants have been identified, such as the rs1256049 (+1082G/A); it is characterized by a change of G by A in exon 5 and its function remains unknown. The rs4986938 (+1730G/A) variant, located in the 3′ untranslated sequence of exon 8, is characterized by a silent transition G > A, their A allele correlates with a lower expression of the ESR2 gene, and the rs1256030 variant is characterized by a transversion of C by T located in the intron 2 (Haldosén, Zhao & Dahlman-Wright, 2014; Mahdavipour et al., 2017; Zhang et al., 2018). Studies of associations between polymorphisms in ESR2 and BC risk have been inconclusive. Some of them have observed associations (Gold et al., 2004; Tsezou et al., 2008; Haldosén, Zhao & Dahlman-Wright, 2014), while others have not (Försti et al., 2003; Al-Eitan et al., 2019). However, in the Mexican population, the association of ESR2 variants (rs1256049, rs4986938, and rs1256030) in BC remains unknown. Thus, the aim of this investigation was to determine the frequency and association of ESR2 variants (rs1256049, rs4986938, and rs1256030) in Mexican women with BC risk.

Materials and Methods

Blood samples from 472 healthy blood-donating women and 400 patients with clinically and histologically confirmed BC were included in the study. All samples for the study group were obtained after the patients and controls provided written informed consent, as approved by the ethical local committee (1,305 and 785) registered under number R-2016-1305-4 of Centro de Investigación Biomédica de Occidente (CIBO), Instituto Mexicano del Seguro Social (IMSS). All procedures performed in studies involving human participants were in accordance with the Helsinki Declaration. Clinical and demographical data were obtained using written questionnaires. New samples were included in the study, as well as stored samples from previous studies, in both cases a code was assigned to maintain the confidentiality of the study participants.

DNA was extracted from peripheral blood lymphocytes using standard protocols (Miller, Dykes & Polesky, 1988).

The PCR amplification of the rs1256049, rs4986938, and rs1256030 ESR2 variants was achieved followed by restriction enzyme analysis according to Table 1 and Fig. 1.

Figure 1 Electrophoretic running on 6% polyacrylamide gel (29:1).

(A) Rsa I digestion. Genotypes GG (321 bp), AA (211 + 110 bp). (B) Alu I digestion. Genotypes GG (442 bp) and AA (336 + 106 bp). (C) Alu I digestion. Genotypes CC (227 + 24 bp), TC (227 + 147 + 80 + 24 bp) and TT (147 + 80 + 24 bp). M represents 50 bp DNA ladder.

Table 1 PCR conditions of rs1256049, rs4986938, rs1256030 polymorphisms in the ESR2 gene.

	rs1256049 (+1082G/A, exon 5)	rs4986938 (+1730G/A, 3′UTR, exon 8)	rs1256030T/C (Intron 2)	
*PCR Primers: sense
antisense	5′-TTGCGCAGCTTAACTTCAAA-3′
5′-ACCTGTCCAGAACAAGATCT-3′	5′-CAATGCATATCCTGCCTGTG-3′
5′-GGTTTAGGGGTGGGGTAGACTG-3′	5′-CAATGCATATCCTGCCTGTG-3′
5′-TCCCGGAAATCTGATACAGC-3′	
PCR product size	321 bp	442 bp	251 bp	
PCR conditions	Buffer enzyme 1X, 7.5 pmol of each primer, 0.2 mM of dNTPs, 2.0 mM of MgCl2, 2.5 u of Taq polymerase, 100 nM of genomic DNA	Buffer enzyme 1X, 7.5 pmol of each primer, 0.2 mM of dNTPs, 2.0 mM of MgCl2, 2.5 u of Taq polymerase, 100 nM of genomic DNA	Buffer enzyme 1X, 7.5 pmol of each primer, 0.2 mM of dNTPs, 2.5 mM of MgCl2, 2.5 u of Taq polymerase, 100 nM of genomic DNA	
Annealing PCR temperature	60 °C	55 °C	57 °C	
Recognition enzyme restriction	RsaI, 37 °C, (5′…GT↓AC…3′)	AluI, 37 °C, (5′…AG↓CT…3′)	AluI, 37 °C, (5′…AG↓CT…3′)	
**Allele identified:
(Wild type)
(Polymorphic type)	G: 321
A: 211 + 110	G: 442
A: 336 + 106	C: 227 + 24
T: 147 + 80 + 24	
Notes:

* The primers previously described (Mahdavipour et al., 2017).

** The genotypes were identified in 6% polyacrylamide gels (29:1), followed by silver nitrate staining.

Although these polymorphisms have been extensively studied in other parts of the world, in few studies their association with BC has been reflected and given that these polymorphisms have been little studied in the Mexican population, we consider it important to explore them. However, the selected polymorphisms such as rs1256049 has very low MAF values in Mexican population, for the above the size of the analyzed sample may be one of the limitations of the study.

The sample size was calculated with a power of 80% and the alpha type error of 0.001 was 168, 342 and 194 cases for polymorphisms rs1256049, rs1256030, and rs4986938 of the ERS2 gene, respectively, using the online genetic power calculator program (https://zzz.bwh.harvard.edu/gpc/) (Purcell, Cherny & Sham, 2003).

Allele frequencies of each variant were obtained by direct counting. The Hardy–Weinberg equilibrium (EHW) was tested by the chi-square goodness-of-fit test to compare the observed genotype frequencies with the expected frequencies among control subjects. Odds ratios and 95% confidence intervals (CI) were also calculated. A two-tailed p < 0.05 was considered statistically significant. The association analysis was determined by odds ratio, and binary logistic regression was performed using PASW Statistic Base 18 software, 2009 (Chicago, IL, USA). Pair-wise linkage disequilibrium (D′) and haplotype frequency were analyzed by the SHEsis Online Version program (Shi & He, 2005).

Results

Table 2 shows the epidemiological data from the BC patients and control individuals. The observed average age in BC patients was 51.45 years, ranging from 23 to 84 years of age, and both tobacco and alcohol consumption were statistically different in BC patients and controls (p < 0.0001).

Table 2 Demographic data for the study groups.

	BC patients(n=400)	Controls(n=472)	p-value	
Age (years)	
Mean (SD)	51.45	(11.86)	50.99	(12.18)	0.09*	
<50 years [(n), %]	(215)	54.0	(256)	54.0		
≥50 years [(n), %]	(185)	46.0	(216)	46.0	0.939**	
Tobacco consumption	
No [(n), %]	(292)	73.0	(423)	90.0		
Yes [(n), %]	(108)	27.0	(49)	10.0	<0.0001**	
Alcohol consumption	
No [(n), %]	(288)	72.0	(430)	91.0		
Yes [(n), %]	(112)	28.0	(42)	9.0	<0.0001**	
Notes:

SD (standard deviation).

* Student’s T-test.

** Chi-square test.

The genotype AA frequency of the rs1256049G/A and rs4986938G/A variants did not show significant differences between patients and controls. However, in the rs1256030C/T polymorphism, the genotype TT (OR 1.86, 95% CI [1.05–3.28], p = 0.042), allele T (OR 1.37, 95% CI [1.09–1.73], p = 0.007), and the recessive model (OR 1.40, 95% CI [1.05–1.87], p = 0.027) were observed as BC risk factors, respectively. The dominant model (OR 0.66, 95% CI [0.49–0.89], p = 0.007) and allele C (OR 0.72, 95% CI [0.57–0.91], p = 0.007) show a protective factor, respectively. The genotype distribution of the rs1256049, rs4986938, and rs1256030 variants in the ESR2 gene were in the Hardy–Weinberg equilibrium in the control group (Table 3A).

a Table 3A Genotype and allelic distribution of the rs1256049G/A, rs4986938G/A, rs1256030T/C variants of ESR2 in BC patients and controls. Frequency comparison of the rs1256049, rs4986938 and rs1256030 ESR2 gene polymorphisms in woman.

Variant	BC	Controls*	OR	95% (CI)	p-value	
rs1256049	Genotype	(n = 400)	%	(n = 324)	%				
	GG	(391)	98	(322)	99	1			
	GA	(9)	2	(2)	1	3.70	[0.79–17.2]	0.123	
	AA	(0)	0	(0)	0				
Dominant	GG	(391)	98	(322)	99	0.27	[0.05–1.25]	0.123	
	GA+AA	(9)	2	(2)	1				
Recessive	AA	(0)	0	(0)	0	1.23	[0.07–19.8]	1.0	
	GG+GA	(400)	100	(324)	100				
	Allele (2n800)			(2n=648)					
	G	(791)	0.988	(646)	0.996	0.272	[0.05–1.26]	0.124	
	A	(9)	0.012	(2)	0.004	3.67	[0.79–17.6]	0.124	
rs4986938	Genotype	(n = 400)	%	(n = 447)	%				
	GG	(300)	75	(347)	78	1			
	GA	(96)	24	(93)	21	1.20	[0.86–1.66]	0.301	
	AA	(4)	1	(7)	1	0.63	[0.18–2.18]	0.553	
Dominant	GG	(300)	75	(347)	78	0.86	[0.62–1.18]	0.373	
	GA+AA	(100)	25	(100)	22				
Recessive	AA	(4)	1	(7)	1	1.57	[0.45–5.42]	0.553	
	GG+GA	(396)	99	(440)	99				
	Allele (2n800)			(2n=894)					
	G	(696)	0.870	(787)	0.880	0.909	[0.68–1.21]	0.577	
	A	(104)	0.130	(107)	0.120	1.099	[0.82–1.46]	0.577	
rs1256030	Genotype	(n = 400)	%	(n = 346)	%				
	CC	(199)	50	(201)	58	1			
	TC	(162)	40	(126)	37	1.18	[0.88–1.59]	0.285	
	TT	(39)	10	(19)	5	1.86	[1.05–3.28]	0.042	
Dominant	CC	(199)	50.5	(201)	58	0.66	[0.49–0.89]	0.0.007	
	CT+TT	(201)	49.5	(145)	42				
Recessive	TT	(39)	10	(19)	5	1.40	[1.05–1.87]	0.027	
	CC+TC	(361)	90	(327)	95				
	Allele (2n800)			(2n=692)					
	C	(560)	0.700	(528)	0.763	0.724	[0.57–0.91]	0.007	
	T	(240)	0.300	(164)	0.237	1.379	[1.09–1.73]	0.007	
Notes:

OR (odds ratio), CI (confidence intervals), p-value (significant <0.05).

* Hardy–Weinberg equilibrium in controls for rs1256049 (chi-square test = 0.0031, p = 0.9555), rs4986938 (chi-square test = 0.071; p = 0.7887), and rs1256030 (chi-square test = 0.016; p = 0.8974). From sample total of controls (n = 472) only were obtained the genotypes for rs1256049G/A (n = 324), rs4986938G/A (n = 447), rs1256030T/C (n = 346) of ESR2 variants.

Comparison of the variants (rs1256049, rs4986938, and rs1256030) in Mexican woman controls with some woman control populations is shown in Table 3B.

Table 3B Frequency comparison of the rs1256049, rs4986938 and rs1256030 ESR2 gene polymorphisms in Mexican women control with other populations.

Polymorphism	Genotypes p-value	
			GG	GA	AA		GG	GA	AA	
rs1256049	
Race	Reference	Country								
Mestizo	(this study)	Mexico	322	2	0					
						vs.				
Caucasian	(Dai et al., 2014)	Germany	5,086	389	5		0.00001	0.00001	0.332	
Asian	(Dai et al., 2014)	Brazil	230	208	29		0.00001	0.00001	0.00001	
Caucasian	(Dai et al., 2014)	USA	6,751	734	70		0.00001	0.00001	0.371	
Caucasian	(Dai et al., 2014)	Sweden	356	41	1		0.00001	0.00001	1.0	
Caucasian	(Dai et al., 2014)	Finland	198	39	1		0.00001	0.00001	0.577	
Asian	(Dai et al., 2014)	China	537	541	131		0.00001	0.00001	0.00001	
rs4986938	
Mestizo	(this study)	Mexico	347	93	7					
						vs.				
African	(Li et al., 2019)	Tunisia	94	120	69		0.00001	0.00001	0.00001	
Caucasian	(Li et al., 2019)	Brazil	109	115	29		0.00001	0.00001	0.00001	
Caucasian	(Li et al., 2019)	Germany	2,169	2,557	752		0.00001	0.00001	0.00001	
Asian	(Li et al., 2019)	Japan	281	102	5		0.097	0.074	0.964	
Caucasian	(Li et al., 2019)	Japan	236	171	51		0.00001	0.00001	0.00001	
Asian	(Li et al., 2019)	India	9	81	159		0.00001	0.0008	0.00001	
Caucasian	(Li et al., 2019)	USA	3,229	3,304	984		0.00001	0.00001	0.00001	
Caucasian	(Li et al., 2019)	USA	470	612	190		0.00001	0.00001	0.00001	
Caucasian	(Li et al., 2019)	Sweden	175	190	56		0.00001	0.00001	0.00001	
Caucasian	(Li et al., 2019)	Sweden	105	103	30		0.00001	0.00001	0.00001	
Caucasian	(Lidaka et al., 2021)	Russia	10	49	8		0.00001	0.00001	0.00001	
Caucasian	(Marini et al., 2012)	Italy	31	45	13		0.00001	0.00001	0.00001	
rs1256030			TT	TC	CC		TT	TC	CC	
Mestizo	(this study)	Mexico	9	126	201					
						vs.				
African	(Greendale et al., 2006)	USA	12	134	147		0.444	0.0443	0.0188	
Caucasian	(Greendale et al., 2006)	USA	126	343	220		0.00001	0.0002	0.00001	
Asian	(Greendale et al., 2006)	USA-Chinese	10	73	67		0.0654	0.0269	0.0026	
Asian	(Greendale et al., 2006)	USA-Japanese	27	75	57		0.00001	0.0514	0.00001	

The comparative analysis of the association between BC patients and controls, stratified by demographic data (age <50 years old, non-smoking, and non-drinking), shows statistically significant differences with the rs1256030 variant (Table 4), while the rs1256049 and rs4986938 variants did not show significant differences between BC patients and controls (data not shown).

Table 4 Association of ESR2 rs1256030 variant with age, non-drinking and non-smoking in the BC patients and controls.

Variant	Genotype	Variable	OR	95% (CI)	p-value	
rs1256030	TT	<50 years old	1.85	[1.05–3.27]	0.043	
	TT	Non-smoking	2.03	[0.70–0.29]	0.018	
	TT	Non-drinking	2.06	[0.72–0.29]	0.019	
	TCTT	Non-smoking	1.40	[0.34–0.15]	0.026	
	TCTT	Non-drinking	1.39	[0.39–0.15]	0.036	
Note:

OR (odds ratio), CI (confidence intervals), p-value (significant <0.05).

The association of clinical characteristics with the ESR2 (rs4986938, rs1256030) variant in the BC group is shown in Table 5. In BC patients who were overweight, the genotype GG of polymorphism rs4986938 was a protector factor (OR 0.21, 95% CI [0.05–0.85], p = 0.043), whereas the GA genotype was a risk factor (OR 1.76, 95% CI [0.56–0.24], p = 0.027) in Luminal A patients. The genotypes TC, TT, and TCTT of the rs1256030 variant were a risk factor in BC patients stratified by stage IV (TC, OR 1.6, 95% CI [1.06–2.54], p = 0.033), Luminal A (TC and TCTT, OR 1.6, 95% CI [0.47–0.21], p = 0.040), and lymph node metastatic (TT, OR 2.67, 95% CI [1.29–5.53], p = 0.010). Overweight BC patients had elevated Ki-67 with the TT genotype (OR 1.67, 95% CI [0.85–3.28], p = 0.041) and Luminal A with ≤4 gestations (TCTT, OR 2.64, 95% CI [1.13–6.14], p = 0.036). However, the TT genotype in the HER2 histological type with lymph node positivity was a protective factor.

Table 5 Association of ESR2 rs1256049, rs1256030 variants with clinical variables of BC patients.

Variant	Histological status	Genotype	Clinical variable*	OR	95% (CI)	p-value	
rs4986938G/A		GG	Overweigh (25–29.9 BMI)	0.21	[0.05–0.85]	0.043	
		GA	Luminal A	1.76	[0.56–0.24]	0.027	
rs1256030T/C		TC	Stage IV	1.6	[1.06–2.54]	0.033	
		TC	Luminal A	1.6	[0.47–0.21]	0.041	
		TT	lymph node metastatic	2.67	[1.29–5.53]	0.010	
		TCTT	Luminal A	1.6	[0.47–0.21]	0.040	
	Ki-67 (≥20%)	TT	Overweigh	1.67	[0.85–3.28]	0.041	
	Luminal A	TCTT	Gestations (≤4)	2.64	[1.13–6.14]	0.036	
	HER2	TT	lymph node metastatic	0.38	[0.18–0.78]	0.005	
Notes:

OR (odds ratio), CI (confidence intervals), p-value (significant <0.05).

* Non-significance clinical variables included in the analysis: Age (<50, ≥50 years), tobacco and alcohol consumption, menopause, cancer type (ductal, lobular), Chemotherapy (non-chemotherapy response), non-chemotherapy response by recurrence. The non-response to chemotherapy treatment with Anthracyclines (e.g. doxorubicin, epirubicin, liposomal doxorubicin), taxanes (docetaxel, paclitaxel) and trastuzumab was evaluated according to the pathological Ryan’s classification described as follows: 1. Moderate response (single cells or small groups of cancerous cells), 2. Minimum response (residual cancer surrounded by fibrosis), and 3. Poor response (minimal or no tumor destruction, extensive residual cancer). Molecular classification (Luminal B, Triple negative).

The rs1256049 and rs1256030 variants were found to be a strong linkage disequilibrium (D′ = 1.0 and r′ = 1.0) in the control group (Fig. 2).

Figure 2 Linkage disequilibrium of the rs1256049 and rs1256030 polymorphisms of ESR2.

Genes were found to be in strong linkage disequilibrium (D′ = 1.0 and r′ = 1.0). The rs4986938 variant was moderately LD of rs1256049 variant.

The haplotype and alleles combination frequency comparisons among the study groups were statistically significantly different in the GAT (OR 3.1, 95% CI [1.31–7.72], p = 0.011), GGC (OR 0.7, 95% CI [0.60–0.97], p = 0.034), and alleles combination GGGATC (OR 2.5, 95% CI [1.28–5.0], p = 0.008), respectively (Tables 6 and 7). In addition, the comparison between alcohol consumption with GGGGTC in BC patients and controls was statistically significantly different (OR 2.6, 95% CI [1.19–5.7], p = 0.022) (data not shown). The association of the alleles combination GGGGCC and GGGGTC with clinical characteristics of BC patients, stratified by molecular classification, are shown in Table 8.

Table 6 Haplotype frequency of the rs1256049, rs4986938, rs1256030 variants of ESR2 in BC patients and controls.

ESR2 gene	BC (n = 800)	Controls (n = 472)			
rs1256049G/A	rs4986938G/A	rs1256030T/C	n	%	n	%	OR 95% (CI)	p-value	
A	G	C	5	0.6	1	0.2	2.9 [0.34–25.4]	0.53	
A	G	T	2	0.3	1	0.2	1.1 [0.10–13.0]	1.0	
G	A	C	72	9.0	46	9.6	0.9 [0.62–1.35]	0.73	
G	A	T	30	4.0	6	1.0	3.1 [1.31–7.72]	0.01	
G	G	C	487	61.0	316	67.0	0.7 [0.60–0.97]	0.03	
G	G	T	203	25.0	102	22.0	1.2 [0.94–1.61]	0.14	
A	A	C	1	0.1	0	0.0			
A	A	T	0	0.0	0	0.0			

Table 7 Alleles combination distribution of ESR2 variants rs1256049, rs4986938, rs1256030 of in study groups.

ESR2 gene	BC (n = 400)	Controls (n = 234)			
rs1256049G/A	rs4986938G/A	rs1256030T/C	n	%	n	%	OR (95% CI)	p-value	
GG	GG	CC	152	38.0	105	45.0	0.7 [0.55–1.06]	0.133	
GG	GG	TC	110	27.5	66	28.0	0.9 [0.66–1.35]	0.843	
GG	GG	TT	32	8.0	12	5.0	1.5 [0.80–3.14]	0.241	
GA	GG	CC	4	1.0	1	0.5	2.3 [0.25–20.9]	0.657	
GA	GG	TC	1	0.25	1	0.5	0.57 [0.03–9.3]	1.0	
GA	GG	TT	1	0.25	0	0.0	1.1 [0.10–12.7]	1.0	
GA	GA	CC	1	0.25	0	0.0	1.1 [0.10–12.7]	1.0	
GA	GA	TC	2	0.50	0	0.0	1.7 [0.17–16.7]	1.0	
GG	GA	CC	43	10.75	32	15.0	0.74 [0.4–1.22]	0.301	
GG	GA	TC	45	11.25	11	5.0	2.5 [1.28–5.0]	0.008	
GG	GA	TT	5	1.25	3	1.0	0.96 [0.22–4.0]	1.0	
GG	AA	CC	2	0.50	3	0	1.7 [0.17–16.7]	1.0	
GG	AA	TC	2	0.50	0	0.0	1.7 [0.17–16.7]	1.0	

Table 8 Association of alleles combination with clinical variables of BC patients group.

Haplogenotype	Histological status	Clinical variable*	OR	95% (CI)	p-value	
GGGGCC	Luminal A	Partial response	0.25	[0.07–0.07]	0.011	
	HER2	Gastrict toxicity	0.21	[0.06–0.74]	0.016	
GGGGTC	Triple negative	Stage I–II	0.3	[0.10–0.87]	0.044	
Notes:

OR (odds ratio), CI (confidence intervals), p-value (significant <0.05).

* Non-significance clinical variables included in the analysis: Age (<50, ≥50 years), tobacco and alcohol consumption, menopause, cancer type (ductal, lobular), Chemotherapy (non-chemotherapy response), non-chemotherapy response by recurrence. The non-response to chemotherapy treatment with Anthracyclines (e.g. doxorubicin, epirubicin, liposomal doxorubicin), taxanes (docetaxel, paclitaxel) and trastuzumab was evaluated according to the pathological Ryan’s classification described as follows: 1. Moderate response (single cells or small groups of cancerous cells), 2. Minimum response (residual cancer surrounded by fibrosis), and 3. Poor response (minimal or no tumor destruction, extensive residual cancer). Molecular classification (Luminal B).

Discussion

The incidence of BC in Mexico has increased in the last 10 years and, as in other parts of the world, is currently one of the principal causes of death in women (Chávarri et al., 2012; Gallegos et al., 2017; Carrillo et al., 2019). In fact, it has been observed that it occurs at an average age of 50 years (Gallegos et al., 2017; Carrillo et al., 2019); the data has been consistent in this study with the average patient age of 51.45 ± 11.86 years. New diagnostic techniques and changes in health politics have contributed to better knowledge about BC, as well as to improving the quality of life of BC patients in our country (Gallegos et al., 2017). However, as BC is a multifactorial disease, it is necessary to implement new studies and strategies for the detection of the disease in its early stages. In this study, we observed the differences in tobacco and alcohol consumption between BC patients and controls. The relationship between these two factors and cancer development is well established (Hydes et al., 2019). It is known that estrogen stimulates mitotic activity in ductal and lobular breast cells, contributing to an increase in cancer risk (Maguire et al., 2005; Mahdavipour et al., 2017). The ESR1 and ESR2 genes function as ligand-activated transcription factors that are stimulated by hormones, translocate to the nucleus, bind to their DNA response elements, and form transcriptional complexes with coactivators, which increases the response of these receptors to regulate the transcription. ERα increases and ERβ decreases the proliferation of cells (Harsløf et al., 2010). Many relevant studies in BC have been associated with different variants of the ESR2 gene, including the rs1256049, rs4986938, and rs1256030 variants (Försti et al., 2003; Maguire et al., 2005; Tsezou et al., 2008; Yu et al., 2011; Karlsson et al., 2016; Haldosén, Zhao & Dahlman-Wright, 2014; Mahdavipour et al., 2017; Ghali et al., 2018). The rs4986938 and rs1256049 are the most extensively studied variants of the ESR2 gene (Yu et al., 2011; Haldosén, Zhao & Dahlman-Wright, 2014), but they have demonstrated contradictory results; some have been associated with susceptibility development to BC (Dai et al., 2014), whereas others have not (Wang et al., 2013; Ghali et al., 2018).

Moreover, little is known about the association of the rs1256049, rs4986938, and rs1256030 variants in Mexican BC patients. In our study group, the frequency of genotypes, alleles, and recessive models of both the rs1256049 and rs4986938 variants were not significantly different between the BC patients and controls (p > 0.05). This data is in accordance with a study of a Swedish population with a total of 723 BC cases and 480 controls, which did not find a statistically significant association for any of the single polymorphisms shown for the rs1256049 variant (Maguire et al., 2005). However, one of the reasons for not observing an association between the rs125049 polymorphism and breast cancer could be the low frequency of the polymorphic allele and probably the sample size analyzed may be insufficient to observe such association.

In our study group, the frequency of rs1256049 and rs4986938 did not represent a risk factor for BC in the Mexican population. These data were consistent with meta-analysis studies that examined different populations and cancer types (Dai et al., 2014; Li et al., 2019). However, when comparing the rs1256049 and rs4986938 genotypes GG and GA of our Mexican women control group with the control groups of other populations, differences were observed with the Asians and Caucasian populations (Greendale et al., 2006). Similarities observed were with respect to the distribution of the AA genotype of the rs1256049 polymorphism in the Caucasian population. In regard to the rs1256030 polymorphism, it showed differences above all with the CC genotype of the African, Caucasian and Asian population compared to the Mexican population. This points to the genetic heterogeneity of this polymorphism in other populations, as well as demonstrating the importance of the sample size used in the different population studies.

Another study that included 436 postmenopausal women with BC from a Han Chinese population found no statistically significant association for the rs4986938 variant (Yu et al., 2011). A meta-analysis study that included both polymorphisms was not associated with BC risk in any model (Breast and Prostate Cancer Cohort Consortium et al., 2008). In addition, we found statistically significant differences with the genotype TT, allele T, and the recessive model CC+TC of the rs1256030 variant (p < 0.05) between BC and control group, observing it as factor of susceptibility risk to the development in BC. This variant has been studied in cognitive development in menopause women (Karlsson et al., 2016), hepatic cirrhosis (Yang et al., 2015), ovarian cancer (Lurie et al., 2009), and BC (Gold et al., 2004; Goulart, Zee & Rexrode, 2009; Al-Eitan et al., 2019). The data are in accordance with a study that demonstrated the association of a haplotype that includes the variant rs1256030 with a susceptibility development risk to BC (Gold et al., 2004). How the rs1256030 variant had recognition sites with a different regulatory sequence may have influenced gene expression or function of ESR2 (Goulart, Zee & Rexrode, 2009); it is possible that the variant might affect their regulation and contribute to tumorigenicity in breast tissue. Furthermore, an intronic mutation may affect splicing sites and lead to a different final protein. In this study, we observed the genotype combinations TC, TT, and TCTT of the rs1256030 variant as a risk factor for susceptibility to BC development in women who are <50 years old, do not drink alcohol, or smoke. Although there are no existing studies that support these findings, these confounding factors show that this stratification is important for contributing to differences in the ESR2 variants’ associations with BC risk. Molecular mechanisms in the regulation of ERβ expression are relatively unclear, probably due to the position of rs1256030 in the promotor region of the ESR2 gene as a recognition site for transcription binding in the regulation of the gene (Goulart, Zee & Rexrode, 2009; Lurie et al., 2009).

We observed that the rs1256049 and rs1256030 variants were risk factors for susceptibility to the development of BC, stratified by the different clinicopathological parameters, such as Luminal A, stage IV, and positive lymph nodes. Although the expression of ERβ in BC has been analyzed in different clinical materials, the regulation mechanisms of ERβ in the development of cancer are still not understood. The expression of ERα and related genes has emerged as one of the major determinants of molecular classification of invasive BC. The correlation of ERβ expression with clinical pathological parameters of BC has been investigated in different studies (Marotti et al., 2010; Rezende et al., 2017). Marotti et al. (2010) studied tissue microarrays of paraffin blocks of 3,093 BC and found a relationship between the expression of ERβ and the molecular class of invasive BC. Overall, 68% of breast carcinomas were ERβ+. The expression of ERβ was significantly associated with the expression of ERα (p < 0.0001) and PR (p < 0.0001). ERβ expression was significantly related to molecular category (p < 0.0001) and was more common in Luminal A (72% of cases) and B (68% of cases). Rezende et al. (2017), observed that rs4986938 sporadically modulates the severity of BC. Zhang et al. (2012), correlated the ERβ with BC stage IV and node positive.

In addition, we observed the protective effect of the TT genotype of the rs1256030 variant in BC HER2 with a positive lymph node. Observations that have been documented by other studies have observed an inverse correlation in both ERβ expression in BC HER2 with positive lymph nodes (Oueslati et al., 2017) and polymorphism in the ESR2 gene (Chattopadhyay et al., 2014). A plausible explanation would be that the rs1256030 variant located upstream in the promoter region has sequence recognition to several transcription factor binding, and the polymorphism might be modified by these binding sites and alter the transcription of ESR1 expression, consequently modulating ESR2 target gene expression (Ghali et al., 2018).

The ESR2 variants were in linkage disequilibrium, and haplotypes were constructed. Two common haplotypes (rs1256049 and rs1256030), comprising approximately 99% of all haplotypes, were found. The GGC haplotype was more frequent in BC patients (61%) and controls (67%) and was associated as a protective factor for susceptibility to BC development. However, the haplotype GGT and alleles combination GGGATT were associated as risk factors for susceptibility to the development of BC. Although the analyzed number of cases for carrying different variations with this allele was low, these can be considered limitations for this study; further, considering the allele frequencies, the MAF value in the study cohort of the rs1256049 variant is very low, and the distribution of alleles within the cohort may be considered insufficient to determine the genotype-phenotype correlation. This limitation may be the main reason why a significant association was not found for this variant.

However, these are a reflection of the common genotypes observed in the Mexican population, no studies exist on these haplotypes analyzed here.

In this study, we also observed, as a protective factor, the association between alleles combination GGGGCC in Luminal A with a partial response, HER2 with gastric (nausea, diarrhea, vomiting, stomatitis, and mucositis), hematological (neutropenia, anemia, and thrombocytopenia) toxicities during chemotherapy treatment, as well as the reduced risk of GGGGTC alleles combination in BC triple negative in stage I–II. Investigations of the expression of ERβ showed that the different molecular subtypes of BC, including Luminal A, Luminal B, HER2, and triple negative, present different profiles of estrogen expression (Mahdavipour et al., 2017). These depend on the moment of therapeutic treatment, before the operation, or after chemotherapy and radiotherapy, which could be a predictive factor for the prognosis of patients with BC (Haldosén, Zhao & Dahlman-Wright, 2014; Yu et al., 2011; Mahdavipour et al., 2017; Carrillo et al., 2019). Moreover, the combined effects of the three polymorphisms in the ESR2 gene might confer a significant genetic predisposition to a complex disease. Thus, a possible explanation for the protective role of polymorphisms in the ESR2 gene could be that the function of ERβ in BC cells displayed different roles depending on the presence of ERα and other ERβ isoforms. When ERα is present, it has anti-proliferative function, but exerts proliferative effects in the absence of ERα. These variations and the different variants in the ESR2 gene, as well as the epigenetic factor, could contribute to the variability of results in clinical studies in BC (Haldosén, Zhao & Dahlman-Wright, 2014; Yu et al., 2016; Mahdavipour et al., 2017; Carrillo et al., 2019).

Although one of the limitations of this study was the size of the sample analyzed by the MAF, there was very low frequency of two of the polymorphisms analyzed. One of the strengths of this study was the strong linkage disequilibrium found between polymorphisms rs1256049 and rs1256030 of the ESR2 gene in control samples. It should be noted that the number of studies available where the LD of these polymorphisms is analyzed is limited.

Conclusions

In conclusion, our results showed that the rs1256030 variant had an association risk factor for BC, comparing controls and BC patients in the genotype TT, allele T, and recessive model (CC+TC genotype). In addition, differences were also observed in patients and controls with regard to carriers of the TT genotype <50 years old, and their non-smoking and drinking status. Thus, the differences are evident in patients with TT, TC, and TCTT genotype with (1) Stage IV, (2) Luminal A, and (3) positive lymph node; in addition, in BC with Ki-67 elevated and overweight with genotype TT and Luminal A with a low number of gestations. The presence of the TT genotype is a protective factor in HER2 patients with positive lymph nodes. The identification of haplotypes included two common factors: the GAT as a risk factor and the GGC as a protective factor. The alleles combination GGGATC as a risk factor and GGGGCC (Luminal A with partial chemotherapy response and HER2 with toxicity gastric), and GGGGTC (triple negative with stage I–II) as protective factors, respectively, confirming that these factors significantly contribute to BC susceptibility in the analyzed sample from the Mexican population.

Further studies are required to confirm these observations, as well as the number of analyzed variations in this study is limited to reflect more extensional results on the alleles combination in a given population other polymorphisms of this gene need to be analyzed in the BC from the Mexican population.

Supplemental Information

Supplemental Information 1 List of variables used to analyze the study data.

Click here for additional data file.

The authors would like to thank the nurses for assisting in the extracting blood samples.

Additional Information and Declarations

Competing Interests

Author Contributions

Human Ethics

Data Availability

The authors declare that they have no competing interests.

Martha Patricia Gallegos-Arreola conceived and designed the experiments, performed the experiments, analyzed the data, prepared figures and/or tables, authored or reviewed drafts of the paper, and approved the final draft.

Guillermo M. Zúñiga-González conceived and designed the experiments, analyzed the data, authored or reviewed drafts of the paper, and approved the final draft.

Luis E. Figuera conceived and designed the experiments, analyzed the data, prepared figures and/or tables, authored or reviewed drafts of the paper, and approved the final draft.

Ana María Puebla-Pérez performed the experiments, analyzed the data, prepared figures and/or tables, authored or reviewed drafts of the paper, and approved the final draft.

María Guadalupe Márquez-Rosales performed the experiments, analyzed the data, prepared figures and/or tables, authored or reviewed drafts of the paper, and approved the final draft.

Belinda Claudia Gómez-Meda performed the experiments, analyzed the data, prepared figures and/or tables, authored or reviewed drafts of the paper, and approved the final draft.

Mónica Alejandra Rosales-Reynoso performed the experiments, analyzed the data, authored or reviewed drafts of the paper, and approved the final draft.

The following information was supplied relating to ethical approvals (i.e., approving body and any reference numbers):

All samples for the study group were 86 obtained after the patients and controls provided written informed consent, as approved by the ethical local committee (1,305 and 785) registered under number R-2016-1305-4 of Centro de Investigación Biomédica de Occidente (CIBO), Instituto Mexicano del Seguro Social (IMSS).

The following information was supplied regarding data availability:

The raw measurements are available in the Supplemental File.

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
