# Peer review of "ESR2 gene variants (rs1256049, rs4986938, and rs1256030) and their association with breast cancer risk"

_PeerJ, doi:10.7717/peerj.13379_

## Round 0.1 · original submission · Minor Revisions

The authors should address all of the issues raised by the reviewers. Especially they need to provide detailed explanation for the comments of reviewer 3.

Reviewer 1 ·

Basic reporting

The manuscript titled as “ESR2 gene variants (rs1256049, rs4986938, and rs1256030) and their association with breast cancer risk” is reviewed and the points that need to be checked are listed below:

-Eventhough the language used in the article in general is clear the terminology used in some parts are not mentioned in a traditional way (for example Line 60-61 bound sites, Line 66-67 cancers express progesterone receptors, line 183-184 cirrhosis biliary, etc) .

-Also in line 46 a nonrelated statement “Add your abstract here” was found in the mansucript.

-In the case of haplogenotype it is strongly suggested to use the regular haplotype or haplogroup terminology. And also in results and discussion part it is better to show this haplogroups by defining the order of the SNPs. For example in line 139 haplotype GGGACT, in here it would be better to show the order of the SNPs and/or only alleles may be included, in general literature it is not common to show it this way. The whole text should be checked for the mentioning of haplogroups/haplotypes.

Experimental design

In materials and methods section in order to understand the method well text may include the full method used in experimental approach. Restriction digestion analysis was not mentioned in the text but in the table. So the explanation of the technique is weak. It would have been also good to see the restriction digestion results on the gel as a figure /figures for each SNP.

Validity of the findings

The aim of the study was stated clearly, there wasn’t any problem with the statistical approach that was used in the text.

·

Basic reporting

Overall, the manuscript is well written with a clear and concise English. The structure, figures and tables are relevant to the text. Mainly the introduction is relevant and almost all publications were referred. The 4 publication I listed below are missing in introduction and discussion parts. It would be better if the authors will refer these articles:
1. Al-Eitan LN, Rababa'h DM, Alghamdi MA, Khasawneh RH. Association between ESR1, ESR2, HER2, UGT1A4, and UGT2B7 polymorphisms and breast Cancer in Jordan: a case-control study. BMC Cancer. 2019 Dec 30;19(1):1257. doi: 10.1186/s12885-019-6490-7.
2. Gallicchio L, Berndt SI, McSorley MA, Newschaffer CJ, Thuita LW, Argani P, Hoffman SC, Helzlsouer KJ. Polymorphisms in estrogen-metabolizing and estrogen receptor genes and the risk of developing breast cancer among a cohort of women with benign breast disease. BMC Cancer. 2006 Jun 29;6:173. doi: 10.1186/1471-2407-6-173.
3. Treeck O, Elemenler E, Kriener C, Horn F, Springwald A, Hartmann A, Ortmann O. Polymorphisms in the promoter region of ESR2 gene and breast cancer susceptibility. J Steroid Biochem Mol Biol. 2009 Apr;114(3-5):207-11.
4. Tsezou A, Tzetis M, Gennatas C, Giannatou E, Pampanos A, Malamis G, Kanavakis E, Kitsiou S. Association of repeat polymorphisms in the estrogen receptors alpha, beta (ESR1, ESR2) and androgen receptor (AR) genes with the occurrence of breast cancer. Breast. 2008 Apr;17(2):159-66.

Experimental design

Even though the overall description of Method and statistical analyses are sufficient, there is a missing point in "Materials&Method" part: It was not clearly indicated which method was used after PCR during genotyping of variants.
Even though the sample size of healthy and patient samples are not low, it would be better if the authors indicate the power analysis of the study.

Validity of the findings

Even though there are many studies that search the association between the ESR2 gene variants and breast cancer in various populations, the manuscript shows this association for the first time in Mexican population.

·

Basic reporting

No comment

Experimental design

The larger population size is always powerful for associaton studies, the studied population size is relatively big enough for this study compared the cohorts with similar studies. The selection criteria of the patients and genotyping methods were well defined and successfully obtained. The selected poplymorphisms and allele frequencies are as important as the size of the population.
The selected polymorphisms such as rs1256049 has very low MAF values in Mexican population . Considering very low number of cases for carrying this allele , population size creates limitation. Therefore, the selection criteria of studied polymorphisms should be explained.

Validity of the findings

Estrogens hormones with their regulating roles in the development of mammary gland and overexpression patterns in breast cancer makes these hormones as a strong candidate for cancer pathogenesis. ESR2 gene have been associated with breast cancer and this gene is a favourable target for these studies.

ESR2 gene polymorphisms and and BC risk have been inconclusive since the regulatory roles of different polymorphic variants on the expression of ESR2 gene is mainly uncertain. Although there are associations, the opposite finding are also available in different studies.
In this study, the research group investigates the association of ESR2 variants (rs1256049, rs4986938, and rs1256030) in breast cancer for Mexican women population with breast cancer risk. Thus, the aim of this investigation was to determine the frequency and association of ESR2 variants (rs1256049, rs4986938, and rs1256030) in Mexican women with breast cancer risk.

For 22 studies including 22,994 cases and 30,514 controls, There was no significant association between rs1256049 and cancer risk in the overall population but not in Asians whereas negative results were obtained for breast cancer in any genetic model.

Considering the allele frequencies, the MAF value in the study cohort of the rs1256049 variant appears to be very low. Therefore, the distribution of alleles within the cohort may be considered insufficient to determine the genotype-phenotype correlation. This limitation may be the main reason why a significant association was not found for this variant.

Authors found that rs1256049 and rs1256030 were found to be strong linkage disequilibrium in the control samples. The available number of cases for carrying different variations with this allele is very limited and LD studies should be reconsidered for this variant. This could have been highlighted in the article.
The application of haplotype and haplogenotype analysis in the study was a very good approach. However, the ESR2 gene is highly polymorphic with 720 SNPs (www.snpper.chip.org). There are many studies of association of ESR2 gene variants with breast cancer in the literature. Other variations located close to the LD could have been taken into account when performing the haplotype and haplogenotype analysis. The number of analyzed variations maybe limited to reflect more extensional results on the haplogenotype in a given population.

Different studies were also conducted in different population cohorts regarding the variations within the scope of the study. Conflicting results have been obtained regarding its association with breast cancer. This study conducted in the Mexican population should have been compared with the data from other individuals on a population basis. The similarities and differences should be discussed in the article and comparative analysis methods may useful to reveal population based differences.

---

## Round 0.2 · accepted · Accept

The manuscript is now ready to be published.